# Relevance of Endoscopic Ultrasound in Endocrinology Today: Multiple Endocrine Neoplasia Type 1, Insulinoma, Primary Aldosteronism—An Expert’s Perspective Based on Three Decades of Scientific and Clinical Experience

**DOI:** 10.3390/cancers15133494

**Published:** 2023-07-04

**Authors:** Peter Herbert Kann

**Affiliations:** 1German Center for Endocrine Care DEVZ, Düsseldorfer Strasse 1-7, 60329 Frankfurt am Main, Germany; kannp@med.uni-marburg.de; 2Faculty of Medicine, Philipp’s University, Baldinger Strasse, 35033 Marburg, Germany

**Keywords:** endocrinology, endoscopic ultrasound (EUS), pancreas, multiple endocrine neoplasia type 1 (MEN1), insulinoma, neuroendocrine tumor, adrenal, adenoma, primary aldosteronism

## Abstract

**Simple Summary:**

In endocrinology, endoscopic ultrasound is a useful and reliable diagnostic technique in the assessment of multiple endocrine neoplasia type 1, localizing insulinomas, and identifying aldosterone-producing adenomas.

**Abstract:**

In endocrinology, endoscopic ultrasound (EUS) may be used to image the adrenals, the endocrine pancreas, and other organs where endocrine neoplasms may occur. During the recent decades, EUS has been established predominantly to assess multiple endocrine neoplasia type 1, to localize insulinomas, and to identify aldosterone-producing adenomas. EUS in endocrinology requires special skills and individual experience in order to provide reliable diagnostic information.

## 1. Introduction

Stimulated by the successful improvement of ultrasound imaging of the ovaries in females using transvaginal ultrasound, endocrine researches tried to reduce the distance between ultrasound transducers and the adrenals and the endocrine pancreas by using endoscopic ultrasound (EUS) in the early nineties of the last century. Using instruments primarily developed for imaging in gastroenterology, it is now possible to generate ultrasound imaging of the retroperitoneal endocrine organs from the stomach and duodenum in higher quality and resolution than ever before. A then newly developed technical approach solved the problem even of imaging the right adrenal [1].

In our center, EUS achieved spectacular diagnostic successes in rare and specific cases such a detecting a 7 mm ACTH-secreting paraesophageal neuroendocrine tumor and proving diagnosis by fine-needle aspiration biopsy, correct localization of an intracardial paraganglioma, and selective blood sampling from the portal vein via fine-needle punction from the stomach in order to localize hormone secretion.

However, looking at three decades experience with EUS in endocrinology, three indications have been established as clinically and quantitatively relevant (Table 1):

-assessment of the pancreatic and adrenal manifestations of multiple endocrine neoplasia type 1 (MEN1);-localization of insulinoma including planning adequate surgical strategy;-detection and localization or even exclusion of aldosterone-producing adenomas (APA, Conn’s adenomas) and defining adequate therapeutic strategy.

Computed tomography (CT) now provides reliable criteria for benign adrenal adenomas, and it turned out that these adenomas usually do not grow in a relevant way [2], and long-term follow-up of small lesions with CT-criteria of benignancy is no longer considered mandatory [3]. Furthermore, clear and reliable criteria to distinguish various entities of adrenal masses using EUS could not be established [4,5,6,7]. Thus, efforts for diagnostic clarification and even follow-up of adrenal “incidentalomas“ became less important and cannot be considered an unambiguous indication for adrenal imaging by EUS anymore.

Gastrinomas and their regional metastases—especially when localized in the gastroduoneal ligament—can be localized correctly by EUS. However, this is a rare indication for EUS in endocrinology because these patients usually present at gastroenterologists. EUS has an encouraging sensitivity for pancreatic gastrinomas. However, the probability of successful localization of gastrinomas in the duodenal wall is disappointing [8,9].

This article aims to provide a concise review of the three indications for EUS in endocrinology considered relevant today.

## 2. Multiple Endocrine Neoplasia Type 1

MEN1 is an autosomal dominant hereditary disease caused by a pathogenic variant in the MEN1 gene on chromosome 11q13.1 [10,11,12]. It is characterized by the occurrence of multifocal neuroendocrine neoplasias (NEN) in two or more endocrine organs, typically in the endocrine pancreas, the pituitary, the parathyroid glands, and the duodenum. Further manifestations include adrenal tumors, neuroendocrine tumors of the thymus and lung as well as lipomas and angiofibromas [13,14,15].

EUS has been established as a useful instrument in MEN1 concerning diagnostic imaging and follow-up assessment [16,17].

We learned that EUS enables detection of even small pancreatic NEN (pNEN) in MEN1 with a diameter of 2 mm and even less [18]. In comparison to other imaging methods, it could be shown that EUS is the most sensitive imaging procedure for the detection of small (≤10 mm) pNEN [19,20].

Regular monitoring of growth velocity of pNENs in MEN1 has been considered mandatory to identify pNENs requiring surgical treatment [21,22]. Therefore, methodical limitations of EUS in terms of the reproducibility of tumor size measurement have been assessed. Based on the variation of tumor diameter (pNENs < 15 mm in MEN1) in repetitive measurements by EUS, the least significant change (*p* < 0.05) could be defined as 15%. The least significant change depended as expected on tumor size and was found for lesions <5 mm somewhat higher with barely 20%, for lesions >5 mm 12% [23].

Subsequent studies revealed that the vast majority of small MEN1 pNENs grow slowly with a growth velocity of approximately 1% per month or even less referring to tumor diameter reflecting a benign behavior [16,24,25,26,27,28].

It is the challenge in clinical follow-up assessment of MEN1 patients to identify the small subset of lesions that reveal criteria of malignant behavior referring to EUS morphology and growth velocity [18,21] (Figure 1).

Recently, data of EUS imaging in a large cohort of MEN1 patients revealed interesting data: pNENs in MEN1 were found to be unequally distributed between the pancreatic anlagen. In the ventral anlage, prevalence and incidence were much lower than in the dorsal anlage, whereas growth velocity of MEN1 pNENs in both anlagen was found to be identical [28,29]. At this time, these findings require confirmation by other groups before surgical consequences—for example, duodenum and ventral pancreas preserving subtotal pancreatectomy [30]—may be considered.

Even if also malignant adrenal tumors may occur in MEN1 [31], adrenal involvement in MEN1 seems to be less relevant than pancreaticoduodenal manifestation. MEN1 associated adrenal tumors are mostly small, benign, and nonfunctioning. However, EUS imaging revealed that they are much more common than previously reported, and EUS has been reported to be the most sensitive imaging procedure [31]. In general, benign adrenal lesions in MEN1 remain unchanged in their morphology during follow-up [15] and usually do not require intervention.

## 3. Insulinoma

The diagnosis of insulinoma is based on positive fasting test and negative serum and urine toxicology for insulin secretagogues (sulfonylureas, glinides) [32,33,34]. Imaging procedures can localize the tumor and, thus, may be relevant for planning the appropriate treatment [35]. especially in the cases where a minimally invasive surgical approach might be appropriate [17,36,37].

EUS has been established as a valuable tool for insulinoma localization and definition of the surgical strategy [36,38]. It has been reported to be more reliable than other imaging techniques [20]. In experienced hands, its diagnostic accuracy may achieve 100% [39].

A risk to miss an insulinoma may be especially relevant in young, slim females where the pancreas in general can be rather hypoechoic and, thus, poorly contrasts the usually hypoechoic insulinoma [37].

Being aware that also falsely positive findings by EUS have been reported, i.e., irregular parenchymatous pancreatic lesions not presenting tumors, for example (post-)inflammatory lesions, cystic lesions, or atypical ventral anlagen [40], it is helpful to know that diagnostic specificity can be improved by combining EUS with fine-needle aspiration biopsy. Even in small tumors, a high success rate can be achieved as a result [35]. This may especially be useful in situations where endocrine testing reveals borderline results. Diagnostic outcome may be further improved by using newer end-cutting fine-needle biopsy needles [41].

Our own data obtained from 65 insulinoma patients revealed that an insulinoma may cause clinically relevant symptoms leading to a diagnosis when it has a diameter of at least 6 mm measured by EUS (“critical cell mass”). This is important to know, especially in situations where an insulinoma is suspected, and various lesions in the pancreas are detected by imaging, as for example in MEN1. Homogenous tumor echogenicity seems to be an indicator for higher differentiation related to more severe clinical symptoms [42].

Recently, EUS-guided ethanol ablation and radiofrequency ablation and have been suggested as alternatives to surgery for the treatment of insulinomas in selected cases [43].

## 4. Primary Aldosteronism

The diagnosis of primary aldosteronism (PA) [44] is based on endocrine screening and confirmation tests as described extensively in a guideline accepted worldwide [45]. In case of confirmed PA, the most relevant diagnostic challenge is to identify the patient that may be cured from her/his arterial hypertension (and chronic hypokalemia) by minimal invasive surgery. Therefore, idiopathic hyperaldosteronism (IHA) needs to be distinguished from APA (Conn’s adenoma) because only the latter usually benefits from surgical treatment, i.e., adenomectomy or adrenalectomy [46].

Traditionally, selective arterial venous sampling has been recommended to identify the source of autonomous aldosterone secretion [45]. However, clinical experience reveals that this is a difficult diagnostic approach requiring special skills, yet it frequently does not provide useful diagnostic information. Documented methodical success rates of AVS from approximately 30% to 70% in terms of reliably obtaining blood samples from both adrenal veins are rather underwhelming [47]. Furthermore, complications may occur, and radiation exposure is not negligible.

An imaging-based diagnostic approach using CT bas been reported to achieve appropriate diagnostic decisions [48].

Casuistic reports suggest that EUS may detect and localize small APAs when CT and/or magnetic resonance imaging (MRI) remained negative [17,49,50,51].

There is just one study published as a brief summary available, yet that study systematically compares EUS to other methods that aim to identify and localize aldosterone-producing adenomas correctly such as CT, MRI, conventional sonography, selective adrenal venous sampling, and posture simulation test—the latter just to differentiate correctly between aldosterone-producing adenoma and idiopathic hyperaldosteronism [52], which will be depicted more in detail here.

In that retrospective study, a total of 112 patients (48 [43%] females, 64 [57%] males, 54 ± 12 years) presenting consecutively for EUS imaging of the adrenals between 1997 and 2019 were analyzed. They fulfilled the two following inclusion criteria:Positive screening/case finding test for PA, i.e., positive aldosterone/renin ratio (A/R) [45], and high serum/plasma aldosterone concentration as suggested by Young [53]; or in early times, in which the A/R had not been established, an elevated aldosterone level with renin concentration in the low normal range was accepted. If the analysis had been performed in our laboratory (supine blood sampling after 30 min bed rest), cutoff for A/R had been defined as 3.8, and high aldosterone serum concentration as ≥ 15.0 ng/dL [53]. If aldosterone and/or renin analysis had not been performed in our laboratory, and the units used had not matched with our methods, cutoffs referring to the respective methods were used.Diagnosis of arterial hypertension, i.e., documented systolic blood pressure ≥ 140 mmHg and/or diastolic blood pressure ≥ 90 mmHg [54], or prevalent antihypertensive medication before EUS imaging.

In the further diagnostic workup, confirmation tests (saline infusion test [SIT], captopril challenge test [CCT], aldosterone and aldosterone metabolites in 24 h urine sampling) and localization tests (EUS, CT, MRI, conventional sonography [CS], AVS, posture stimulation test [PST]) were performed.

### 4.1. Saline Infusion Test (SIT)

SIT has been established as the standard confirmation test during the study period. It was performed based on a recommendation by the German Society for Endocrinology [55]. Patients stayed in the recumbent position for at least 8 h before and during the continuous i.v. infusion of 2 l of 0.9% NaCl solution over 4 h, starting between 8.00 and 9.00 in the morning. Blood samples for renin (baseline), aldosterone, cortisol, potassium, and sodium (all at 0–2–4 h) were taken. Different cutoffs for aldosterone (5 ng/dL; 10 ng/dL) were in discussion and used over time. Referring to data from Giachetti et al. [56] and Rossi et al. [57], the Endocrine Society Clinical Practice Guideline [45] suggests a cutoff of 6.8 ng/dL for offering the best trade-off between sensitivity and specificity. For this study, all SIT results were reassessed according to an aldosterone cutoff of 6.8 ng/dL after 4 h. The test result was also considered normal when baseline A/R was found to be normal.

### 4.2. Captopril Challenge Test (CCT)

CCT has been suggested as another confirmation test during the study period. It was performed and interpreted based on a recommendation by the German Society for Endocrinology [55] and a suggestion by the Endocrine Society Clinical Practice Guideline [45]. Patients stayed in the recumbent position for at least 8 h before and during the test. They received 50 mg captopril at baseline. Blood samples for renin, aldosterone, and potassium at 0–1–2 h were taken. The test result was considered normal when aldosterone could be suppressed >30%.

### 4.3. Aldosterone and Aldosterone Metabolites in 24 h Urine Sampling

Following a suggestion by Mourad et al. [58], a 24 h urinary aldosterone secretion <12 µg was considered normal. If metabolites such as tetrahydroaldosterone or 18-aldosterone glucuronide were assessed, the cutoff for the respective method was applied.

### 4.4. Computed Tomography (CT); Magnetic Resonance Imaging (MRI); Conventional Sonography (CS)

Results from CT/MRI/CS as available in written reports from our hospital and performed elsewhere were included in the analysis. CT/MRI/CS techniques, imaging procedures, and documentations had not been standardized throughout the study period.

### 4.5. Selective Adrenal Vein Sampling (AVS)

Data obtained from AVS have been accepted for our study, if catheterization of the adrenal veins via a percutaneous femoral vein approach and blood sampling from these locations and a cubital vein or the infrarenal inferior caval vein, respectively, have been documented [45]. Procedures, technical details, and documentation were not standardized throughout the study period. Successful catheterization was defined by a selectivity index, i.e., adrenal versus peripheral/infrarenal vein cortisol ratio > 2 without cosynthropin use [59]. A lateralization index aldosterone/cortisol dominant versus non-dominant side >2 was considered defining lateralization [59].

### 4.6. Posture Stimulation Test (PST)

PST has been suggested to provide some evidence to differentiate between APA and IHA. It was performed based on a recommendation by the German Society for Endocrinology [55]. Patients stayed in the recumbent position for at least 8 h before the test. After drawing the baseline blood sample, they stood in an upright position for 3 h. Blood samples for renin and aldosterone at 0 and 3 h were taken. In case of test abruption due to orthostatic symptoms, final blood samples were taken at the time of 0 abruption. An increase in serum aldosterone ≥30% of baseline value was considered to be indicative for IHA, whereas an increase of <30% or a decrease was indicative for APA [60]. If A/R was found to be normal at baseline of PST, PST was interpreted as not confirming PA and was not used to differentiate between APA and IHA.

### 4.7. Endoscopic Ultrasound (EUS)

EUS was carried out according to a standard procedure described previously [17] by one single investigator (phk).

### 4.8. Definition of Aldosterone-Producing Adenoma (APA)

APA was defined based on a reassessment of the endocrine parameters, histopathology, and criteria of therapeutic success following suggestions by Vorselaars et al. [61].

### 4.9. Data Analysis

Planning a retrospective diagnostic study on PA and performing a reliable data analysis is a challenge. Just to cite the Endocrine Society Clinical Practice Guideline [45]: “The current literature does not identify a ‘gold standard’ confirmatory test for PA. […] There is empirical evidence that case-control designs for establishing the accuracy of diagnostic tests overestimate their accuracy.”

Furthermore, at least slightly different protocols for tests and imaging procedures—which of course also may change over time—are used in different centers. Finally, it needs to be stated that even surgical treatment does not yield a final diagnostic gold standard because while the pathologist can determine that there is an adrenocortical adenoma, performing immunohistochemistry for CYP11B2 (aldosterone synthase) [62] is not a diagnostic standard. Thus, not even the pathologist can tell whether an adrenal adenoma found in the preparation has in fact been secreting aldosterone.

Defining IHA is more problematic still. Its diagnosis is merely based on an agreement on the interpretation of laboratory results, which themselves are prone to methodical imprecision and other influencing factors, in combination with missing evidence for APA by imaging and lateralization tests. Usually, histomorphological findings are not available, since the affected patients do not undergo surgical treatment. Even therapeutic success of the use of mineralocorticoid antagonists is no evidence for the correct diagnosis, since these are known to lower blood pressure and to increase serum potassium regardless of potential causes of arterial hypertension or/and hypokalemia.

Regarding these preconditions, it is a priori challenging to design a study on the accuracy of any diagnostic approach in PA.

The aim of this study is exclusively to investigate whether EUS might be considered a helpful diagnostic tool in identifying and localizing an APA. Therefore, any attempt to compare any diagnostic methods in order to reveal which method might be better and which be worse is deliberately excluded.

In this study, patients with APA have been defined stepwise according to the following criteria:

Step 1: total study cohort

Step 2: exclusion of patients with unlikely PA

If there was not at least one positive confirmation test (SIT, CCT, aldosterone, or aldosterone metabolites in 24 h urine sampling) or indication for prevalent adenoma by at least one test (MRI, CT, sonography, AVS, PST), PA was considered unlikely. These patients were excluded from further analysis.

Step 3: post hoc definition of patients with APA

Definition criteria for this group were

-diagnosis of adrenal adenoma by histopathology;-preoperatively serum or plasma aldosterone highly normal or above the normal range as described above with○elevated A/R;○or serum or plasma renin below the normal range or in the very low normal range (in early times when A/R has not been applied).
-≥50% of the following criteria as available needed to be fulfilled:○positive SIT or CCT;○elevated aldosterone or aldosterone metabolites in 24 h urine sampling;○positive lateralization in AVS;○documented postsurgical clinical improvement following suggestions by Vorselaars et al. [61], for this study defined by postoperative arterial normotension without antihypertensive treatment or reduced antihypertensive treatment;○postsurgical normalization of aldosterone and renin in serum or plasma.


#### 4.9.1. Major Study Endpoint

The results of EUS (and the other localization tests) regarding the patients with APA have been analyzed in terms of correct identification and localization in a descriptive manner.

#### 4.9.2. Minor Study Endpoints

(a) The applicability of typical morphological criteria for adrenal adenomas (non-functioning and subclinical Cushing’s syndrome) [2] on APA was studied.

(b) Further patients that underwent surgery for suspected APA but did not fulfill the APA criteria in our post hoc analysis were analyzed in a descriptive manner.

## 5. Results

Out of the total cohort of 112 patients included in the study, 44 patients were defined as APA.

Major study endpoint: EUS detected and localized 39/44 (89%) of APAs correctly. Their largest diameter determined by EUS was 15.6 ± 5.4 mm (range 5.7–26.0 mm; lateralization: left n = 27 [61%]—right n = 17 [39%]).

In four other cases (9%), an unclear/untypical finding was seen by EUS and documented in the examination report (in three cases: “possible adenoma”, in one case: “tumor not typical for an adenoma”). In one case (2%), EUS revealed a slight, unspecific thickening of the organ. In 38 cases (86%), the contralateral adrenal was found to be free of any adenoma-suspicious or nodular lesions. In four cases, EUS showed an additional adrenal adenoma in the contralateral adrenal gland. In two cases, additional adenomas of the same side were found. In three cases, nodular formations of unclear type (i.e., not typical for an adenoma) were detected in the contralateral gland (n = 2) or in the same adrenal gland (n = 1).

In the four cases of bilateral adenomas, the hypersecreting tumor was identified with the help of AVS in two cases. In one case with unsuccessful AVS, it was decided pragmatically to operate the larger right-sided adenoma. In another case, AVS could not be performed because of a contrast medium allergy. At the time of diagnosis, one left-sided adenoma was prevalent, and two others were detected during a follow-up time of three years. The adenoma prevalent at the time of diagnosis was operated.

In our study, EUS was the only method localizing the APA correctly in eight cases (18%). Their largest diameter determined by EUS was 14.9 ± 4.6 mm (range 8.0–23.0 mm). Here, CT was performed in three, MRI in four, and CS in two cases, and it did not reveal an adenoma. In one case, EUS was the only imaging performed and AVS was not diagnostic in this case.

In 26 CTs, APA was described and localized correctly in 21 (81%) cases. By MRI, APA was described and localized correctly in 19/25 patients (76%). Using conventional sonography, adenoma-typical findings have been documented in 8/19 cases. However, two of them were in the contralateral adrenal. Thus, the adenoma was localized correctly in 6/19 (32%) cases.

AVS had been performed in 14 patients. In five patients, a selectivity index > 2 for both sides could be obtained. In all these five AVS, the lateralization index was >2, hence positive for correct localization. Thus, AVS was diagnostic in 36%.

Twenty-one PSTs could be analyzed. In 14 (67%) tests, a result considered typical of IHA had been reported, and in 5 (24%) cases, the result was indicative for APA. Correct test outcomes had been confirmed in five APA patients. Among those was one patient who had been tested twice, one test showing an IHA-typical result. Twelve patients with typical PST results for IHA underwent surgery referring to other testing and were finally found to have APA (including again one patient being double-tested) (Table 2).

Minor study endpoints:

(a) For this analysis, the 39 EUS findings of adenomas as well as the 5 findings considered unspecific—post hoc classified as APA—were analyzed. Typical morphological criteria by EUS of APA were hypoechoic echogenicity, largely homogenous echostructure, capsular tumor margin, termination of medullary echo at tumor margin, largest diameter <2 cm, missing or just slight hyperperfusion by duplex ultrasound. The frequency of APAs fulfilling these criteria is given in Table 3.

APAs tended to be more homogenous, hypoechoic, and smaller compared to non-functioning adrenal adenomas and those causing subclinical Cushing’s syndrome [2].

(b) Another six patients underwent surgery for confirmed PA and preoperatively diagnosed APA. However, they did not fulfill the post hoc definition criteria of APA in our analysis either due to missing diagnostic testing criteria (n = 3, histologically confirmed adenomas) or due to missing histopathological verification of an adenoma (n = 3, histological examination showing nodular-hyperplastic changes in the adrenals). In these latter three “falsely positive” cases, AVS revealed a positive lateralization in all of them. Thus, it seems possible that these were aldosterone-producing cell clusters, APCCs, which contain a high amount of the enzyme aldosterone synthetase and are thought to be precursor tumors of APA, as they are found in normal adrenal glands as well as in patients with PA. Their metabolic phenotype shows great similarities with the ones of APA [62].

No EUS-associated complications were observed.

## 6. Discussion

This study has some limitations that need to be mentioned in this context. As primary aldosteronism is a rare disease, this study has been performed retrospectively and covers a time period of 22 years. It included 112 patients with confirmed PA, with 44 APAs referring to endocrine parameters, histopathology, and criteria of therapeutic success [61]. Another point of criticism may be that all EUS examinations had been performed by one ambitious investigator. Data from other methods and tests included in this study were derived from more or less standardized procedures. Concerning CT, MRI, selective adrenal venous sampling, and confirmation tests, data sets were partly incomplete, and laboratory methods had been changed over the course of the study, which lasted for more than two decades. EUS had not been performed blinded to other test results, because the primary investigator of the study was not merely the one performing all EUS procedures, but rather also responsible for all clinical and therapeutic decisions.

Based on this study and the data available yet, it cannot definitively be concluded that EUS would be superior to any other method in detecting APAs. However, it seems reasonable to state that EUS can be useful when detecting and localizing APAs and also when defining the contralateral adrenal as morphologically normal, especially in cases where CT and/or MRI are negative, and selective adrenal venous sampling—if performed—is methodically unsuccessful.

In a case of clearly confirmed PA, evidence for APA in one adrenal (typical example: Figure 2) and a completely normal morphology of the other adrenal by EUS may be taken as sufficient information to define an indication for surgery without any other localization tests. However, it would be desirable to confirm this suggestion by a prospective randomized trial.

## 7. Conclusions

EUS may be very useful in some rare, very special indications, when atypically located endocrine tumors need to be localized.

In clinical routine, EUS is a useful and reliable diagnostic technique in endocrinology, predominantly to assess multiple endocrine neoplasia type 1, to localize insulinomas, and to identify aldosterone-producing adenomas.

However, EUS in endocrinology requires highly specialized skills and individual experience of the examiner in order to provide reliable diagnostic information and, thus, is not available everywhere.

## Figures and Tables

**Figure 1 cancers-15-03494-f001:**
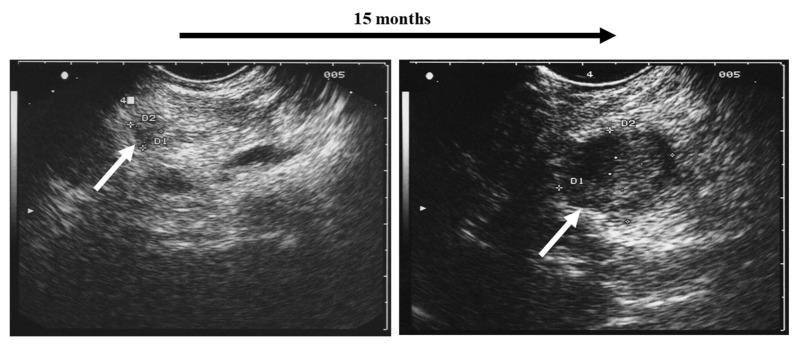
Rapidly growing pancreatic tumor (white arrows) in a MEN1 patient detected by regular annual screening: surgical resection revealed malignancy (neuroendocrine carcinoma) with a small regional lymph node metastasis.

**Figure 2 cancers-15-03494-f002:**
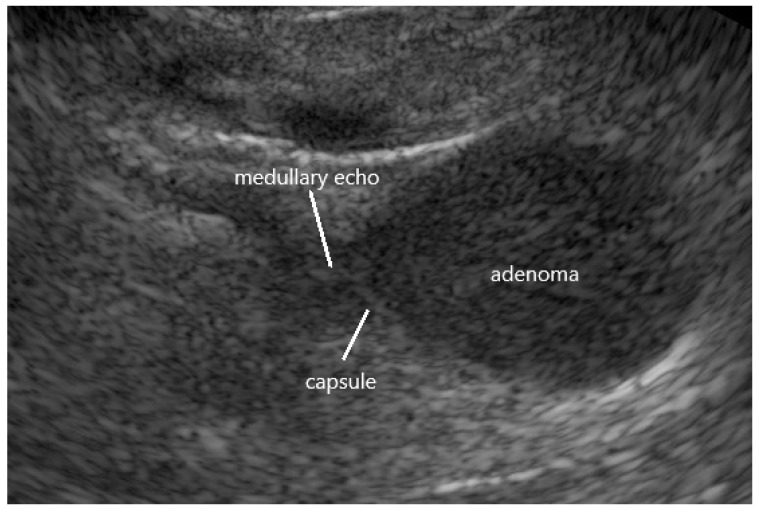
Endoscopic ultrasound of a typical aldosterone-producing (Conn’s) adenoma (APA) in the left adrenal hypoechoic echogenicity, homogenous echostructure, hyperechoic capsular tumor margin, termination of medullary echo at tumor margin.

**Table 1 cancers-15-03494-t001:** Endoscopic Ultrasound in Endocrinology: Indications and Diagnostic Impact.

Multiple Endocrine Neoplasia Type 1	Detection of small tumors from 1–2 mm in diameter → defining “baseline morphology” of each lesion for follow-up. Precise assessment of tumor growth velocity (least significant change < 20% even in very small tumors). Detection of suspicious tumor morphology, morphological changes in follow-up. Detection of atypically fast growing tumors. Planning of surgical intervention: topographical relations of tumors to critical structures; defining parts of the pancreas that should be removed/may be retained; detection of neighbored lymph nodes with suspicious morphology; defining surgical strategy: enucleation, distal resection, minimal invasive surgery, robot-assistance.
Insulinoma	Localization of a pancreatic lesion typical of an insulinoma: hypoechic, round or slightly oval, homogenous or slightly heterogenous benign: clearly defined margin, <2 cm; becoming symptomatic if diameter ≥6 mm; no hyperperfusion. Confirming diagnosis by fine-needle aspiration biopsy. Planning of surgical intervention: topographical relations to critical structures; defining surgical strategy: enucleation, distal resection, minimal invasive surgery, robot-assistance.
Primary Aldosteronism (Conn’s Adenoma)	Identification of an adrenal lesion with typical morphological features of an aldosteronoma. Defining the remaining parts of the ipsilateral and the contralateral adrenal as morphologically inconspicuous/normal. Planning of surgical intervention: selective adenomectomy possible?

**Table 2 cancers-15-03494-t002:** Correct localization of Conn’s adenomas (aldosteronoma) by different methods referring to 44 aldosterone-producing adenomas (APA) proven by laboratory testing and postoperative histology [51]. EUS: endoscopic ultrasound; CT: computed tomography; MRI: magnetic resonance imaging; CS: conventional sonography; SAVS: selective adrenal venous sampling; PST: posture stimulation test.

EUS	89%	39/44	in four cases bilateral adenomas detected
CT	81%	21/26	
MRT	76%	19/25	+one falsely located in the contralateral adrenal
CS	32%	6/19	+two falsely located in the contralateral adrenal
SAVS	36%	5/14	selectivity index * <2 in 9 cases
PST	24%	5/21	referring to correct identification APA vs. IHA

* successful catheterization of the adrenal veins was defined by selectivity index, i.e., adrenal versus peripheral/infrarenal vein cortisol ratio > 2 without cosynthropin use [59] (adapted from [52]).

**Table 3 cancers-15-03494-t003:** Typical morphology of Conn’s adenomas (aldosteronoma) by endoscopic ultrasound referring to 44 aldosterone-producing adenomas proven by laboratory testing and postoperative histology (adapted from [52]).

hypoechoic echogenicity	82%
largely homogenous echostructure	72%
hyperechoic capsular tumor margin	93%
termination of medullary echo at tumor margin	97%
diameter ≤ 2 cm	77%
no/only discrete hyperperfusion (duplex)	94%
tumors fulfilling at least four criteria	98%
tumors fulfilling at least five criteria	91%

## Data Availability

Not applicable.

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
