# Peer review of "Relevance of Endoscopic Ultrasound in Endocrinology Today: Multiple Endocrine Neoplasia Type 1, Insulinoma, Primary Aldosteronism—An Expert’s Perspective Based on Three Decades of Scientific and Clinical Experience"

_cancers, 2023, doi:10.3390/cancers15133494_

Round 1

Reviewer 1 Report

Very interesting and comprehensive review. My comments:

Some tables summarizing the available studies in this filed could be useful to the reader.

The author should comment more on the potential role of EUS-RFA in insulinomas, commenting the results of the recent comparative study PMID: 36871765)

The authors should comment on the paradigm shift in tissue sampling of these lesions due to the use of newer end-cutting FNB needles (in this regard cite the recent NMA PMID: 35124072)

Author Response

Dear reviewer,

thank you very much for considering my manuscript a very interesting and comprehensive review!

Thank you for your suggestion to include tables summarizing available studies in this field. I have done this in a previous publication (Rev Endocr Metab Disord 2018;19:133-137) which was a systematic review. This manuscript, however, has been designed a „perspective“, that I wrote at the end of my academic and clinical career. Therefore, I decided to provide a summary of my suggested indications for endoscopic ultrasound in endocrinology and their clinical relevance in table 1 in this manuscript.

You raise the very interesting point of a potential role of EUS-RFA in insulinomas. We have got in Marburg a really good endocrine surgery (Prof. Rothmund, Prof. Bartsch, Prof. Holzer), maybe one of the best in the world, providing high quality minimal invasive surgical treatments and outcomes. It this environment, there was no need – end even no opportunity for me – to develop alternative therapeutic strategies to treat insulinomas for example using EUS-RFA. Thus – and to come to the point – I have no personal experience with this technology. Because this manuscript is a „perspective“ and not a systematic review, I decided not to include a chapter on technologies where I do not have any personal experiences. Nevertheless, I noted briefly the possibility to treat insulinomas with ethanol ablation or RFA in lines 128/129 and cite the paper you suggested now as reference „45“. I would be really happy of you might please accept this. However, if you might consider this a mandatary condition for publication of my paper, it would of course not be a problem to include an additional chapter in this manuscript. In this case – please tell me!

Concerning your suggestion to mention the possibilities to improve the quality of EUS tissue sampling using newer end-cutting FNB needles, an according statement has been included (line 121/122).

Thank you very much for stating that the use of the English language is fine. Nevertheless, because the other reviewers suggested to improve the quality of the paper by giving it to a native speaker for correction and improvement, this has been performed.

Reviewer 2 Report

The presentation of the current paper as a expert’s perspective is coherent with points 1, 2, 3 and 5.

Under point 4, a retrospective review of the literature is presented, with, in my opinion, slightly insufficient level of detail concerning research design and methods.

The readability might be better if the paper would be presented as a literature review of PA and the discussion would be used to present the results of this review in the larger context of other NEN’s.

examples of formulations that can be improved: lines 93, 124, 363

Author Response

Dear reviewer,

thank you very much for spending time to read and to evaluate my manuscript.

Your point of criticism is that this is a retrospective review of the literature with slightly insufficient level of detail concerning research design and methods.

Please allow me to state here that the type of this paper is a „perspective“ and not a systematic review. Recently, I published a systematic review comparing different methods of imaging NENs, mainly focussing on a comparison of EUS and MRT, but also addressing other imaging methods. In this manuscript, I clearly defined the criteria for literature research and including publications in this systematic review. Main criteria were sufficient study size and availability of postoperative histology as golden standard (Rev Endocr Metab Disord 2018;19:133-137). I cited this paper in my perspective as reference „20“ were a interested reader may find these more detailed information. 

This manuscript, however, has been designed a „perspective“, that I wrote at the end of my academic and clinical career. This perspective provides a somewhat more subjective point of view, nevertheless of course referring to published evidence.

It would really be nice to be able to present a literature review on EUS in PA, of course! Unfortunately, there is not a sufficient number of reports on the use of EUS in PA available in literature. Furthermore, most of these papers have been published be myself or (former) members of my group. You may believe it or not: The study that I present here is the only one available with an acceptable study size.

Concerning more details of PA in terms of pathogenesis, diagnostic procedures and therapeutic option, I refer to the worldwide accepted guideline by Funder et al., reference „47“.

Concerning the quality of English language, also reviewer 3 raised concerns. Therefore, this manuscript has now been corrected by a native speaker.

Reviewer 3 Report

General:  This is a manuscript describing endocrinologist’s perspective on the use/relevance of EUS in assessing endocrinologically active lesions.

Major issues:

1.     Role of EUS in identification of endocrinologically pertinent active lesions are depicted in this article. Inclusion of more recent developments in contrast-enhanced EUS could be incorporated for evaluation of such lesions. 

2.     Comparison of EUS to nuclear studies such as PET-11C-metomidate or DOTATATE scans for localization of tumors are reported and should be discussed. Possibly enhancing/ complementing the finding of neuroendocrine lesion such as gastrinoma in Passaro’s (gastrinoma) triangle, among others.

3.     In initial evaluation of adrenal mass with EUS prior to verification of endocrine related tumors – caveat for FNAs need to be noted if considered – which are performed after exclusion of potential for pheochromocytoma – e.g., in ruling out suspected or occult metastasis.

Minor issues:

- See Language Quality section.

- Spell & grammar check

Author Response

Dear reviewer,

thank you very much for spending time to read and to evaluate my manuscript.

The type of my paper is a „perspective“ and not a systematic review. My „perspective“ provides a somewhat more subjective endocrinologist’s point of view on the use of EUS in endocrinology, nevertheless of course referring to published evidence.

I absolutely agree with you that more recent developments in contrast-enhanced EUS have a potential to improve diagnostic precision. However, to be honest, I have no sufficient personal experience in using contrast-enhanced EUS and did not publish any data on this yet, and thus decided not to include this into this „perspective“. (Also reviewers 1 and 2 did not require this.) I hope this might be acceptable to you. Nevertheless, if you might consider it a mandatory condition for publication of my „perspective“, it would of course be easy to include a chapter on contrast-enhanced EUS. In this case, please tell me!

I also agree with your statement on the role of nuclear studies in finding neuroendocrine lesions. Recently, I published a systematic review comparing different methods of imaging NENs, mainly focussing on a comparison of EUS and MRT, but also addressing other imaging methods such as nuclear studies. I cited this paper in my perspective as reference „20“ where an interested reader may find these more detailled informations.

Indeed, I plan to write another systematic review next year about recent developments in endocrine imaging where I of course also will address recent developments in nuclear studies.

You are completely right in stating that invasive diagnostic procedures such as EUS-biopsies are dangerous and therefore contraindicated in adrenal lesion if a pheochromcytoma has not been excluded. I pointed this out in several previous publications in EUS in endocrinology and also point this out in lectures that I give on the use of EUS in endocrinology. However, and I hope for your understanding, I do not see the need to address thisissue in the present paper. In the context of diagnostic procedures in PA, there is no indication for EUS-biopsy yet. I hope this might be acceptable to you.

Concerning the quality of English language, also reviewer 2 raised concerns. Therefore, this manuscript has now been corrected by a native speaker (spell and grammar check).

Round 2

Reviewer 1 Report

The revised version of the manuscript is OK. Thank you!

Author Response

Thank you!

Reviewer 3 Report

Please check for error/typo -  e.g., "critical cell mess" - Line 125

Author Response

Thank you very much!